# Physical limits to sensing material properties

Farzan Beroz [1✉], Di Zhou [1], Xiaoming Mao[1] & David K. Lubensky[1]

All materials respond heterogeneously at small scales, which limits what a sensor can learn. Although previous studies have characterized measurement noise arising from thermal fluctuations, the limits imposed by structural heterogeneity have remained unclear. In this paper, we find that the least fractional uncertainty with which a sensor can determine a material constant $\lambda_0$ of an elastic medium is approximately $\delta\lambda_0/\lambda_0 \sim (\Delta_\lambda^{1/2}/\lambda_0)(d/a)^{D/2}(\xi/a)^{D/2}$ for $a \gg d \gg \xi$, $\lambda_0 \gg \Delta_\lambda^{1/2}$, and $D > 1$, where $a$ is the size of the sensor, $d$ is its spatial resolution, $\xi$ is the correlation length of fluctuations in $\lambda_0$, $\Delta_\lambda$ is the local variability of $\lambda_0$, and $D$ is the dimension of the medium. Our results reveal how one can construct devices capable of sensing near these limits, e.g. for medical diagnostics. We use our theoretical framework to estimate the limits of mechanosensing in a biopolymer network, a sensory process involved in cellular behavior, medical diagnostics, and material fabrication.

[1] Department of Physics, University of Michigan, Ann Arbor, MI 48109, USA. ✉email: farzan@umich.edu

A fundamental way of learning about a material is by observing how it responds to external stimuli. The functional dependence of a response on a stimulus is known as a constitutive relation. The most basic example of such a relation is Hooke's law $F = kX$ for the deformation response $X$ of a linear elastic solid to a force stimulus $F$, where $k$ is a material constant that is a characteristic property of the solid[1,2]. This linearity is a generic feature of material response for small enough stimuli, as it requires only that the constitutive relation be analytic and non-vanishing to first order. Linear constitutive relations have proven useful for characterizing a broad range of physical systems, including dielectric materials[3], diffusion[4], friction[5], geomaterials[6], Newtonian fluids[7], piezoelectric materials[8], thermoelectric materials[9], and even abstract entities such as financial markets[10,11].

Material constants of linear constitutive relations are typically inferred by comparing the known value of an applied stimulus to the measured response produced by the stimulus. For the case of a homogeneous elastic solid, the material constant is simply given by $k = F/X$. In reality, however, all materials are heterogeneous on a small enough scales[12–15]. This heterogeneity serves as a source of measurement noise that becomes significant for systems that operate at the microscale, such as miniature electronic devices[16–19], medical microrobots[20–23], and biological sensors[24–30].

Previous studies of sensing in random media have focused on remote sensing or communication via traveling waves[31–36]. The inference of material properties at small scales has been studied in microrheology[37–39] and for chemical sensing[40–43]. In these contexts, the measurement noise due to thermal fluctuations has been characterized using fluctuation–dissipation theorems[41,44]. However, thermal fluctuations are fundamentally different from structural heterogeneities: the former arise uniformly in space and vary in time, and the latter vice versa. Although methods are available to probe heterogeneous materials on small scales, it is not known how precisely this process can be done[13,45–48]. What are the limits to sensing the properties of heterogeneous materials, and how can a physical device be designed to achieve these limits?

To quantify the limits of sensing constitutive relations, we investigate a simple model of a localized sensor that probes a heterogeneous medium to estimate a global material constant. Specifically, we consider a continuous medium with a material constant given by a uniform average value $\lambda_0$ plus a spatially varying fluctuation $\delta\lambda(\mathbf{r})$ with short-ranged correlations. We treat the sensor as a spherical device that can probe $\lambda_0$ by applying an external stimulus field and measuring the resulting response field in equilibrium.

In what follows, we show that this inference process admits an optimal (minimum-variance unbiased) measurement protocol. Surprisingly, the optimal protocol depends qualitatively on both the spatial resolution of the sensor and also on whether it can perform multiple probes. For a single probe, the optimal protocol is remarkably complex, because the modes applied to the medium can interfere with each other in a geometrically frustrated manner, akin to the spins of a spin-glass. In contrast, the optimal protocol for multiple probes is comparatively simpler, because it avoids unnecessary interference effects. We exploit this simplicity to determine the total amount of information that the sensor can extract by probing a given region of the medium. Physically, optimal performance is achieved by decoding the results of a sequence of probes that penetrate into the surrounding medium to varying extents. This strategy can allow the sensor to effectively average the material constant over a volume that grows with the spatial resolution of the sensor. Finally, we use our theoretical framework to bound the precision of mechanosensing in a biopolymer network, a sensory process that regulates cellular

behavior in decisive ways[28,49–51] and is used for medical diagnostics and material fabrication[52–54].

## Results

**Probing a Winkler foundation**. To gain insight into sensing material properties in physical space, we explore a minimal model that consists of a spherical sensor embedded in a heterogeneous medium (Fig. 1). In this section, we start by taking the medium to be the simplest heterogeneous material: a disordered Winkler foundation[55]. This medium corresponds to an array of decoupled springs in the continuum limit. The internal energy of the Winkler foundation is given by:

$$E = \frac{1}{2} \int \lambda(\mathbf{r}) u(\mathbf{r})^2 \, d\mathbf{r}, \qquad (1)$$

where $\lambda(\mathbf{r})$ is a spatially varying material constant and $u(\mathbf{r})$ is the response field at position $\mathbf{r}$. We assume $\lambda(\mathbf{r}) = \lambda_0 + \delta\lambda(\mathbf{r})$, where $\lambda_0$ is a fixed, uniform field and $\delta\lambda(\mathbf{r}) \ll \lambda_0$ is a Gaussian random field with zero mean and spatial correlations given by:

$$\langle \delta\lambda(\mathbf{r})\delta\lambda(\mathbf{r}') \rangle = \frac{\Delta_\lambda}{(2\pi)^{D/2}} e^{-(\mathbf{r}-\mathbf{r}')^2/\xi^2}, \qquad (2)$$

where $\Delta_\lambda \ll \lambda_0^2$ is the local variability of $\lambda(\mathbf{r})$, $D$ is the spatial dimension, and $\xi$ is the correlation length of the fluctuations in $\lambda(\mathbf{r})$. For simplicity, we assume $\xi$ is small enough that these correlations can be approximated by:

$$\langle \delta\lambda(\mathbf{r})\delta\lambda(\mathbf{r}') \rangle = \Delta_\lambda \xi^D \delta(\mathbf{r} - \mathbf{r}'). \qquad (3)$$

The quenched disorder $\delta\lambda(\mathbf{r})$ in the material constant limits the precision with which a physical sensor can infer $\lambda_0$. To determine these limits, we consider an idealized sensor that probes $\lambda_0$ by first applying a stimulus field $f(\mathbf{r})$. This field perturbs the energy of the system as follows:

$$\delta E = -\int f(\mathbf{r}) u(\mathbf{r}) \, d\mathbf{r}. \qquad (4)$$

After applying this stimulus, the sensor measures the response of the medium in equilibrium. In particular, we assume that the sensor records an integrated response $m$:

$$m = \int w(\mathbf{r}) u(\mathbf{r}) \, d\mathbf{r}, \qquad (5)$$

where $w(\mathbf{r})$ is a weight field. Taken together, the probe fields $f(\mathbf{r})$ and $w(\mathbf{r})$ define the measurement protocol of the sensor. For any physical sensor, these fields must be localized in space. We impose this locality by constraining the probe fields to obey $f$

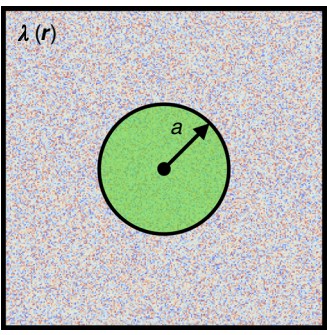

**Fig. 1 Sensing in a heterogeneous medium.** Schematic illustration of sensing model, showing an idealized, spherical sensor of radius $a$ (green) embedded inside a medium with a spatially varying material constant field $\lambda(\mathbf{r})$ (background). The sensor can learn about $\lambda(\mathbf{r})$ by applying an arbitrary stimulus and recording an arbitrary weighted response within its volume.

$(\boldsymbol{r}) = 0$ and $w(\boldsymbol{r}) = 0$ for $r > a$, where $r$ is the radial coordinate and $a$ is the radius of the sensor.

Finally, upon recording the integrated response $m$, the sensor produces an estimate for $\lambda_0$. In what follows, we will determine the optimal estimator $\hat{\lambda}_0$ perturbatively to leading order in $\delta\lambda(\boldsymbol{r})$. In this approximation, the integrated response is:

$$m = \int \left( \frac{1}{\lambda_0} - \frac{\delta\lambda(\boldsymbol{r})}{\lambda_0^2} \right) \psi(\boldsymbol{r}) \mathrm{d}\boldsymbol{r}, \quad (6)$$

where we have defined the probe intensity $\psi(\boldsymbol{r}) \equiv f(\boldsymbol{r})w(\boldsymbol{r})$, a function of the probe fields that captures how strongly the probe senses a given location of space. For a fixed choice of $\psi(\boldsymbol{r})$, along with prior knowledge of the model parameters other than $\lambda_0$, the optimal estimator of $\lambda_0$ based on the outcome of $m$ is (Supplemental Material, Supplementary Note 1):

$$\hat{\lambda}_0 = \frac{s}{m}, \quad (7)$$

where $s$ is a normalizing constant chosen such that the estimator $\hat{\lambda}_0$ yields an unbiased estimate of $\lambda_0$:

$$s = \int \psi(\boldsymbol{r}) \mathrm{d}\boldsymbol{r}. \quad (8)$$

Equation (7) is a mesoscopic generalization of Hooke's law $k = F/X$. By computing the estimate $\hat{\lambda}_0$, the sensor obtains a weighted spatial average of $\lambda(\boldsymbol{r})$:

$$\hat{\lambda}_0 = \frac{\int \psi(\boldsymbol{r})\lambda(\boldsymbol{r}) \mathrm{d}\boldsymbol{r}}{\int \psi(\boldsymbol{r}) \mathrm{d}\boldsymbol{r}}, \quad (9)$$

to leading order in $\delta\lambda(\boldsymbol{r})$. This estimator is optimal in that it has a lower standard deviation $\delta\lambda_0 \equiv \sqrt{\langle (\hat{\lambda}_0 - \lambda_0)^2 \rangle}$ than any other unbiased estimator for a fixed choice of $\psi(\boldsymbol{r})$. Therefore, the optimal measurement protocol can be determined by minimizing $\delta\lambda_0^2$ with respect to the probe intensity $\psi(\boldsymbol{r})$. Inserting Eq. (3) into the definition of the variance yields:

$$\delta\lambda_0^2 = \Delta_\lambda \xi^D \frac{\int \psi(\boldsymbol{r})^2 \mathrm{d}\boldsymbol{r}}{\left( \int \psi(\boldsymbol{r}) \mathrm{d}\boldsymbol{r} \right)^2}. \quad (10)$$

This variance is invariant with respect to an overall rescaling of $\psi(\boldsymbol{r})$. To eliminate this redundancy, we constrain $\int \psi(\boldsymbol{r})\mathrm{d}\boldsymbol{r}$ to be a fixed constant. Furthermore, we must enforce $\psi(\boldsymbol{r}) = 0$ in the exterior of the sensor ($r > a$) to satisfy the constraints imposed by the finite size of the sensor. Thus, the minimum of $\delta\lambda_0^2$ is determined by the configuration of $\psi(\boldsymbol{r})$ that extremizes the following action $S$:

$$S = \int_{\mathcal{R}_{\mathrm{int}}} \left( \frac{1}{2}\psi(\boldsymbol{r})^2 - \gamma\psi(\boldsymbol{r}) \right) \mathrm{d}\boldsymbol{r}, \quad (11)$$

where the integral is taken over the interior $\mathcal{R}_{\mathrm{int}}$ of the sensor ($r < a$) and $\gamma$ is a Lagrange multiplier that fixes $\int \psi(\boldsymbol{r})\mathrm{d}\boldsymbol{r}$. This action is extremized by any measurement protocol with a probe intensity $\psi(\boldsymbol{r})$ that is uniform over $\mathcal{R}_{\mathrm{int}}$. The optimal measurement protocol is, therefore:

$$\psi(\boldsymbol{r}) = \begin{cases} \gamma, & r < a. \\ 0, & r > a. \end{cases} \quad (12)$$

Inserting Eq. (12) into Eq. (10) yields:

$$\delta\lambda_0^2 = \Delta_\lambda \xi^D V^{-1}, \quad (13)$$

where $V$ is the volume of the sensor. Thus, the fractional uncertainty of the estimator $\hat{\lambda}_0$, defined as the standard deviation

$\delta\lambda_0$ divided by the mean $\lambda_0$, scales as:

$$\frac{\delta\lambda_0}{\lambda_0} \sim \left( \frac{\Delta_\lambda}{\lambda_0^2} \right)^{1/2} \left( \frac{\xi}{a} \right)^{D/2}, \quad (14)$$

which can be interpreted as the familiar $1/\sqrt{N}$ scaling of measurement uncertainty for $N$ independent samples. In this analogy, the sample size $N \sim (a/\xi)^D$ corresponds to the number of effectively independent subvolumes probed by the sensor.

**Probing an elastic sheet**. For the Winkler foundation, our model sensor could not induce a response beyond its volume. In contrast, many other types of elastic media are coupled in space and thereby respond to stimuli nonlocally. To understand how such nonlocality affects a sensor's ability to infer material properties, we now turn to conventional, linear elasticity. For simplicity, we will first focus on an isotropic, two-dimensional elastic sheet characterized by a single material constant, and in section "The precision of biomechanical sensing", we will generalize our theoretical framework to a three-dimensional elastic medium characterized by a material constant tensor.

For the elastic sheet, we consider the deformation response $u(\boldsymbol{r})$ to force stimuli $f(\boldsymbol{r})$ oriented perpendicular to the plane of the sheet. Thus, the sheet's internal energy depends on the gradient $\nabla u(\boldsymbol{r})$ of the response field as follows:

$$E = \frac{1}{2}\int \lambda(\boldsymbol{r})\nabla u(\boldsymbol{r}) \cdot \nabla u(\boldsymbol{r}) \mathrm{d}\boldsymbol{r}. \quad (15)$$

Here, as in the previous section, we take $\lambda(\boldsymbol{r})$ to be a Gaussian random field with mean $\lambda_0$, variance $\Delta_\lambda \ll \lambda_0^2$, and spatial correlations over a scale $\xi$. As before, we take the sensor to interact with the medium within a radius $a$ by first applying a stimulus field $f(\boldsymbol{r})$ as in Eq. (4), and then measuring an integrated response $m$ as in Eq. (5).

To leading order in $\delta\lambda(\boldsymbol{r})$, the sensor can again compute $\hat{\lambda}_0 = s/m$ to obtain a spatial average of $\lambda(\boldsymbol{r})$ weighted by a probe intensity $\psi(\boldsymbol{r})$, as in Eq. (9) (Supplemental Material, Supplementary Note 2). However, for the elastic sheet, $\psi(\boldsymbol{r})$ is now:

$$\psi(\boldsymbol{r}) = (\nabla \cdot)^{-1} f(\boldsymbol{r}) \cdot (\nabla \cdot)^{-1} w(\boldsymbol{r}), \quad (16)$$

where $(\nabla \cdot)^{-1}$ is the inverse divergence operator (Supplementary Note 2). This probe intensity is a nonlocal function of the probe fields and thereby allows the sensor to probe distant regions beyond its boundary.

Intuitively, probing a greater extent of the medium should yield a more accurate estimate of $\lambda_0$. To that end, the greatest possible extent of a probe is achieved by probe potentials with a $\sim 1/r$ radial dependence in the far-field limit. For the elastic sheet, this decay profile is not produced by monopoles (which yield pathological, non-decaying potentials), but rather by dipoles. The simplest possible measurement protocol with dipole probe fields is described by:

$$f(\boldsymbol{r}) \sim \delta(r - a)\cos(\theta), \quad (17)$$

$$w(\boldsymbol{r}) \sim \delta(r - a)\cos(\theta). \quad (18)$$

These probe fields cast a probe intensity $\psi(\boldsymbol{r})$ that is uniform in the interior of the sensor and isotropically decaying in the exterior:

$$\psi(\boldsymbol{r}) = \begin{cases} \gamma, & r < a. \\ \gamma\left(\frac{a}{r}\right)^4, & r > a. \end{cases} \quad (19)$$

Inserting Eq. (19) into Supplementary Equation 31 yields the following variance:

$$\delta\lambda_0^2 = \frac{1}{3}\Delta_\lambda \xi^D V^{-1}, \tag{20}$$

for $D = 2$. As expected from dimensional analysis, this expression has the same dependence on the model parameters as for the Winkler foundation (cf. Eq. (13)). Importantly, however, its prefactor is smaller. Thus, our example illustrates how a sensor can harness a long-ranged response function to perform at a higher precision by effectively averaging $\lambda(\boldsymbol{r})$ over a larger region of space.

**Probe-field interference limits the channel capacity of sensing.** Given that a probe of the elastic sheet can access nonlocal information, what limits its precision? To answer this question, we start by considering the simpler case of a sensor that can only apply probe fields on its boundary. For such boundary probes, the most general probe fields are of the form:

$$f(\boldsymbol{r}) \sim \delta(r-a)\sum_k B_k^{(f)} e^{ik\theta}, \tag{21}$$

$$w(\boldsymbol{r}) \sim \delta(r-a)\sum_k B_k^{(w)} e^{-ik\theta}, \tag{22}$$

where $B_k^{(f)}$ and $B_k^{(w)}$ are complex coefficients that satisfy $B_{-k}^{(f)} = B_k^{(f)*}$ and $B_{-k}^{(w)} = B_k^{(w)*}$ to ensure that the probe fields are real, and we assume $k > 0$ to avoid pathological, non-decaying interactions caused by monopoles. This measurement protocol casts the following probe intensity:

$$\psi_{\pm}(\boldsymbol{r}) = \sum_{k,l} B_{kl}(kl + |k||l|)\left(\frac{r}{a}\right)^{\pm|k|\pm|l|-2} e^{i(k-l)\theta}, \tag{23}$$

where $\psi_+(\boldsymbol{r})$ and $\psi_-(\boldsymbol{r})$ are probe intensities that correspond to the interior ($r < a$) and the exterior ($r > a$) of the sensor, respectively, and $B_{kl} \sim B_k^{(f)} B_l^{(w)}$. Inserting $\psi_{\pm}(\boldsymbol{r})$ into the definition of the variance and performing the spatial integrals yields:

$$\delta\lambda_0^2 = \Delta_\lambda \xi^D \sum_{k,l,m,n} B_{kl} B_{mn} \mathbb{T}_{klmn}, \tag{24}$$

where $\mathbb{T}_{klmn}$ is a highly structured, fourth-order tensor:

$$\mathbb{T}_{klmn} = 4\pi a^2 \frac{\delta_{k-l+m-n,0} x_{klmn} y_{klmn}}{(x_{klmn}+2)(x_{klmn}-2)}. \tag{25}$$

Here, $\delta_{i,j}$ is the Kronecker delta function, $x_{klmn} = |k| + |l| + |m| + |n|$, and $y_{klmn} = (kl + |kl|)(mn + |mn|)$. In Eq. (24), we have normalized $\psi_{\pm}(\boldsymbol{r})$ such that $\int \psi(\boldsymbol{r})_{\pm} d\boldsymbol{r} = 1$, which implies that $B_{kl}$ must obey:

$$\sum_k 4\pi a^2 |k| B_{kk} = 1. \tag{26}$$

To gain insight into the optimal measurement protocols for boundary probes, we used the Nelder–Mead algorithm to numerically minimize $\delta\lambda_0/\lambda_0$ over $B_k^{(f)}$ and $B_k^{(w)}$ (see "Methods" section). To that end, we imposed a cutoff on the system by truncating the sums in Eqs. (24) and (26) at a maximum absolute mode number $k_{max}$. Physically, this parameter corresponds to the spatial resolution of the sensor, which we define as:

$$d = \left(\frac{2\pi}{k_{max}}\right)a. \tag{27}$$

For all choices of $k_{max}$, we studied, this algorithm converged to basins of minima dominated by the dipole modes ($k = 1$), which

makes intuitive sense given that these modes probe the largest extent of the medium. Interestingly, however, as we increased $k_{max}$, we found that at certain special values, the optimal probe fields shifted and picked up additional higher-order modes, resulting in a smaller minimum fractional uncertainty $\delta\lambda_{0,\min}/\lambda_0$ (Fig. 2a).

The higher-order modes contribute with smaller amplitudes and nontrivial relative phase shifts (Fig. 2b). These complex configurations arise because different terms in Eq. (24) can provide conflicting contributions to $\delta\lambda_0^2$ depending on the relative phases of the modes. This geometrical frustration greatly suppresses modes beyond the dipole-dipole and quadrupole-quadrupole pairs, which for $2 \le k_{max} \le 12$ appear together with amplitudes and phase relations that maximize the extent of $\psi(\boldsymbol{r})$ while preserving its isotropy. Including three or more mode pairs must break isotropy, analogous to how three or more anti-ferromagnetic spins cannot simultaneously minimize their interaction energies (Supplementary Note 3). Nevertheless, for $k_{max} > 12$, the optimal measurement protocols contain additional higher-order modes that cause small wrinkles in $\psi(\boldsymbol{r})$ (Fig. 2c). Although these wrinkles break the isotropy of $\psi(\boldsymbol{r})$, they also smoothen out its profile in the radial direction, which results in a greater overall uniformity throughout space and thus a higher precision.

To better understand the asymptotic behavior of $\delta\lambda_{0,\min}/\lambda_0$ for large $k_{max}$, we imagine relaxing the constraints on $\delta\lambda_0^2$ by allowing $B_{kl}$ to be an arbitrary matrix satisfying $B_{-k,-l} = B_{kl}^*$. This relaxation expands the space of possible $\psi(\boldsymbol{r})$ to include all real configurations that can be generated by Eq. (23), some of which cannot be cast by a physical probe. Importantly, this relaxation is a convex function of $B_{kl}$, and thus has a unique minimum $\delta\lambda_{0,\text{low}}/\lambda_0$ that provides a theoretical lower bound on $\delta\lambda_0/\lambda_0$. Specifically, in the limit $k_{max} \to \infty$, we find that $\delta\lambda_{0,\text{low}}/\lambda_0 \approx \xi^D \Delta_\lambda V^{-1}/\sqrt{\pi}$, which provides a close lower bound on the values obtained via numerical minimization (Fig. 2a and Supplementary Note 4).

A simple argument based on symmetry reveals that this lower bound must be a strict inequality for $k_{max} > 2$. This argument follows from observing that for all $k_{max}$, the unique optimal configuration of $\psi(\boldsymbol{r})$ for the relaxation is isotropic, in contrast to the configurations we found by minimizing Eq. (24) for $k_{max} > 12$ (Supplementary Note 4). This broken isotropy must persist for all higher values of $k_{max}$, and therefore a boundary probe can never cast a configuration of $\psi(\boldsymbol{r})$ that performs as well as the optimal $\psi(\boldsymbol{r})$ for the convex relaxation of $\delta\lambda_0/\lambda_0$. This example illustrates how interferences between the probe fields limit the information that can be gleaned from a single probe, i.e. the channel capacity of sensing. In the following section, we will show how a sensor can overcome this limit by performing multiple probes, and then we will generalize our results to a sensor that can apply arbitrary probe fields in its volume.

**Sensory multiplexing can significantly improve the precision of sensing.** The interferences in the previous section occur because all of the modes applied by a probe interrogate the medium simultaneously. In principle, however, each mode couples to a different spatial extent of the medium and therefore should carry independent information about $\lambda_0$. Such information could potentially be accessed by performing separate measurements with distinct spectra.

To test this notion, we determine the optimal estimator for a sensor that can perform multiple probes with varying measurement protocols. We label each probe by an integer $k$ and constrain their probe fields to be zero for $r > a$. In this case, the minimum-variance unbiased estimator of $\lambda_0$ is again given by a

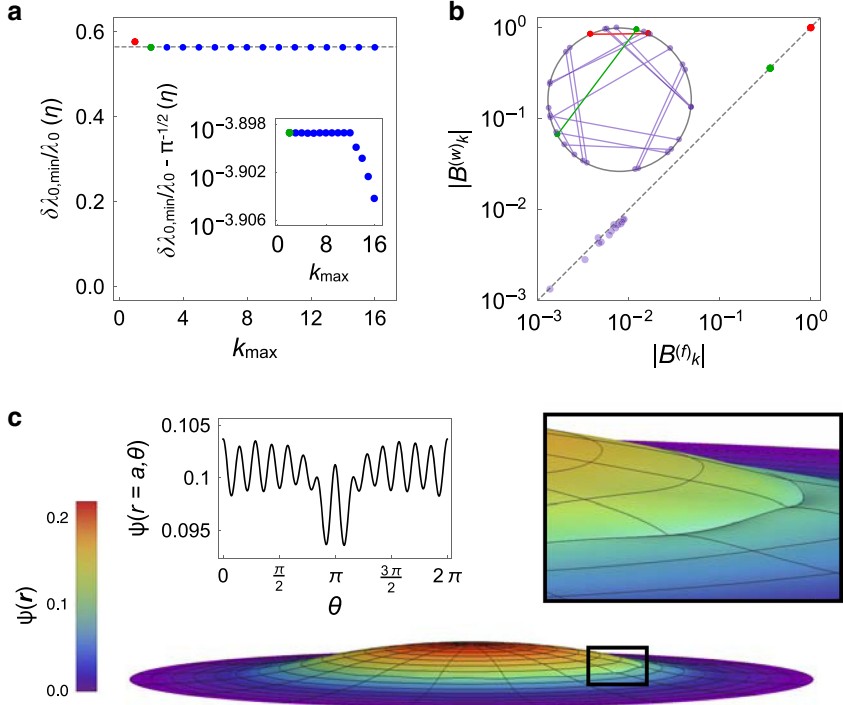

**Fig. 2 Probe-field interference can limit the information that a sensor can glean from a single probe. a** Fractional uncertainty $\delta\lambda_{0,min}/\lambda_0$ in units of $\eta = \xi^D \Delta_\lambda V^{-1}$ for numerically optimal boundary probes versus maximum absolute mode number $k_{max}$ included in the multipole expansions of the probe fields (see section "Probe-field interference limits the channel capacity of sensing"). Red point corresponds to $k_{max} = 1$, green point to $k_{max} = 2$, and blue points to $k_{max} > 2$. Dashed gray line indicates the optimum $\delta\lambda_{0,low}/\lambda_0 \approx \eta/\sqrt{\pi}$ attained in the limit $k_{max} \to \infty$ for the convex relaxation of $\delta\lambda_O/\lambda_0$ described in section "Probe-field interference limits the channel capacity of sensing". Inset: $\delta\lambda_{0,min}/\lambda_0 - \eta/\sqrt{\pi}$ in units of $\eta$ versus $k_{max}$ on a logarithmic scale. **b** Absolute values of the weight field coefficients $|B_k^{(w)}|$ versus the absolute values of the stimulus field coefficients $|B_k^{(f)}|$ for an example measurement protocol obtained via numerical optimization for $k_{max} = 16$, showing dipole modes (red), quadrupole modes (green), and higher-order modes (blue). Dashed gray line shows $|B_k^{(w)}| = |B_k^{(f)}|$. Inset: phases of $B_k^{(f)}$ and $B_k^{(w)}$ for the same measurement protocol as in the main panel. Lines connect the coefficients that correspond to the same value of $k$. Colors same as in the main panel. **c** Probe intensity $\psi(\mathbf{r})$ for the same measurement protocol as in **b** versus spatial coordinate $\mathbf{r}$. Left inset: $\psi(\mathbf{r})$ at the boundary of the sensor ($r = a$) versus angular coordinate $\theta$. Right inset: larger view of the region indicated by the black rectangle in the main panel, showing small wrinkles in $\psi(\mathbf{r})$.

weighted spatial average of $\lambda(\mathbf{r})$:

$$\hat{\lambda}_0 = \frac{\int \Psi(\mathbf{r})\lambda(\mathbf{r})d\mathbf{r}}{\int \Psi(\mathbf{r})d\mathbf{r}}. \tag{28}$$

Here, $\Psi(\mathbf{r})$ is an effective probe intensity created by the optimally weighted sum of the probe intensities $\psi_k(\mathbf{r})$ for the individual probes:

$$\Psi(\mathbf{r}) = \sum_k p_k \frac{\psi_k(\mathbf{r})}{\int \psi_k(\mathbf{r})d\mathbf{r}}, \tag{29}$$

where $p_k = \sum_l C_{kl}^{-1}$ with $C_{kl} \equiv \langle(\hat{\lambda}_{0,k} - \lambda_0)(\hat{\lambda}_{0,l} - \lambda_0)\rangle$ defined as the covariance matrix of the estimators $\hat{\lambda}_{0,k}$ for the individual probes (Supplementary Note 5). The variance of $\hat{\lambda}_0$ is:

$$\delta\lambda_0^2 = \left(\sum_{k,l} C_{kl}^{-1}\right)^{-1}. \tag{30}$$

For simplicity, we start by considering a sensor that applies a sequence of probe fields:

$$f_k(\mathbf{r}) \sim \delta(r - a)\cos(k\theta), \tag{31}$$

$$w_k(\mathbf{r}) \sim \delta(r - a)\cos(k\theta). \tag{32}$$

from an initial mode number $k = 1$ up to a maximum mode number $k = k_{max}$, which corresponds to the spatial resolution $d$ of the sensor defined by Eq. (27). By varying $k$, the sensor

modulates the range of $\psi_k(\mathbf{r})$ in the exterior at the cost of simultaneously modulating $\psi_k(\mathbf{r})$ in the interior:

$$\psi_k(\mathbf{r}) \sim \begin{cases} \left(\frac{r}{a}\right)^{2k-2}, & r < a. \\ \left(\frac{r}{a}\right)^{-2k-2}, & r > a. \end{cases} \tag{33}$$

Interestingly, this collection of boundary probes does not achieve a significant improvement over a single optimal boundary probe. Instead, as $k_{max}$ is increased, the fractional uncertainty approaches $\delta\lambda_0/\lambda_0 \approx \xi^D \Delta_\lambda V^{-1}/\sqrt{\pi}$, as we found for the convex relaxation in section "Probe-field interference limits the channel capacity of sensing". This agreement is not a mere coincidence: for boundary probes, the possible configurations of $\Psi(\mathbf{r})$ are mathematically equivalent to the possible configurations of $\psi(\mathbf{r})$ for the convex relaxation of a single probe (Supplementary Note 6). However, unlike the convex relaxation, the collection of boundary probes reveals an additional physical effect that can limit the precision of a sensor. That is, for multiple probes, the overlapping configurations of $\psi_k(\mathbf{r})$ in the interior correlate the probes and thereby suppress the amount of information that can be extracted from the exterior. These correlations are reflected in the structure of the covariance matrix:

$$C_{kl} = \frac{1}{4}\Delta_\lambda\xi^2 V^{-1}\left(\frac{kl}{k+l-1} + \frac{kl}{k+l+1}\right). \tag{34}$$

In this expression, the first and second fractions are contributed by overlaps in the interior and exterior, respectively.

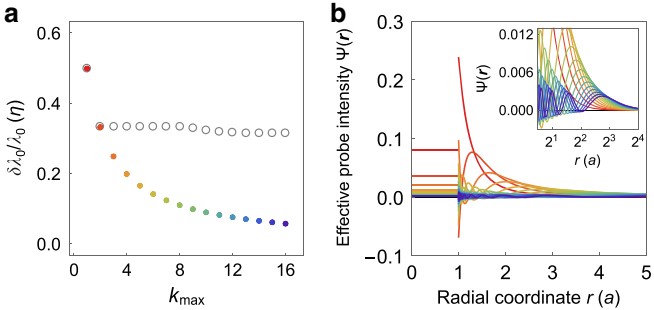

**Fig. 3 Sensory multiplexing can greatly improve the precision of sensing.**
**a** Colored points show the smallest attainable fractional uncertainty $\delta\lambda_0/\lambda_0$ in units of $\eta = \xi^D \Delta_\lambda V^{-1}$ for a sensor that can perform sensory multiplexing up to a maximum absolute mode number $k_{max}$ (see section "Sensory multiplexing can significantly improve the precision of sensing"). Gray circles show a lower bound $\delta\lambda_{0,low}/\lambda_0$ on the fractional uncertainty for each value of $k_{max}$ for a single volume probe, obtained via numerical minimization (Supplementary Note 10). **b** All-inclusive effective probe intensities $\Psi(r)$ for sensory multiplexing versus radial coordinate $r$ in units of the sensor radius $a$ for the same values of $k_{max}$ as in **a** (correspondence indicated by matching colors). Inset shows $\Psi(r)$ on a lin-log scale.

To compensate for the superfluous contributions from the interior, the sensor must employ probe fields that are nonzero within its volume. One way to perform this compensation is by pairing each probe $k$ with a companion probe described by (Supplementary Note 6):

$$\tilde{\psi}_k(\boldsymbol{r}) \sim \begin{cases} \left(\frac{r}{a}\right)^{2k-2}, & r < a. \\ 0, & r > a. \end{cases} \qquad (35)$$

Pairing these companion probes with the original probes using Eq. (29) with appropriate values of $p_k$ yields effective probe intensities $\Psi_k(r)$ that are zero in the interior (Supplementary Note 6):

$$\Psi_k(\boldsymbol{r}) \sim \begin{cases} 0, & r < a. \\ \left(\frac{r}{a}\right)^{-2k-2}, & r > a. \end{cases} \qquad (36)$$

Finally, the sensor may include an additional unpaired companion probe $\tilde{\psi}_k(\boldsymbol{r})$ with $k = 1$ to uniformly sample $\lambda(\boldsymbol{r})$ in its interior. With these adjustments, the resulting all-inclusive effective probe intensity $\Psi(\boldsymbol{r})$ exhaustively decodes the information that can be extracted by probing the region $r < a$ using any combinations of probe fields (Supplementary Note 7). In this case, the covariances among the paired probes $\Psi_k(\boldsymbol{r})$ and the unpaired probe $\tilde{\psi}_1(\boldsymbol{r})$ are given by:

$$C_{kl} = \Delta_\lambda \xi^2 V^{-1}\left(\delta_{k,0}\delta_{l,0} + \frac{kl}{k+l+1}\right), \qquad (37)$$

for $k, l \geq 0$, where $k, l = 0$ correspond to the unpaired probe. Inserting the inverse of this matrix into Eq. (30) yields:

$$\delta\lambda_0^2 = \Delta_\lambda \xi^2 V^{-1}\left(\frac{1}{k_{max}+1}\right)^2. \qquad (38)$$

This variance decreases with $k_{max}$ because each additional probe increases the uniformity of $\Psi(\boldsymbol{r})$ over space (Fig. 3). In the limit of fine resolution $k_{max} \gg 1$ ($d \ll a$), the fractional uncertainty of the sensor's estimate scales as:

$$\frac{\delta\lambda_0}{\lambda_0} \sim \left(\frac{\Delta_\lambda}{\lambda_0^2}\right)^{1/2}\left(\frac{d}{a}\right)^{D/2}\left(\frac{\xi}{a}\right)^{D/2}, \qquad (39)$$

for $D = 2$. Thus, simultaneously varying both probe fields throughout the volume of the sensor can allow a significant

amount of additional information to be transmitted across the sensory channel. We refer to the strategy of performing multiple measurements with varying probe fields as "sensory multiplexing."

Sensory multiplexing can be generalized to $D = 3$ by taking the probe fields to be pairs of spherical harmonics. In this case, $\delta\lambda_0/\lambda_0$ still obeys the asymptotic scaling in Eq. (39) (Supplementary Note 8). Moreover, this scaling is robust to the omission of a finite number of modes (Supplementary Note 9). Taken together, our results reveal that for $d \ll a$, sensory multiplexing can improve the fractional uncertainty of a sensor by a factor proportional to the number $(a/d)^D$ of distinct subvolumes that it can resolve simultaneously for $D = 2$ and $D = 3$.

Notably, this level of precision can never be attained by a single probe, even if the sensor is permitted to apply an arbitrary pair of probe fields within its volume. This limitation occurs due to probe-field interference, as before in section "Probe-field interference limits the channel capacity of sensing". That is, including more than three pairs of boundary modes breaks the isotropy of $\psi(\boldsymbol{r})$, and a sensor can always improve upon an anisotropic $\psi(\boldsymbol{r})$ by performing multiple rotated copies of the probe and combining the results using Eq. (28). Moreover, numerical optimization suggests that Eq. (38) does not provide a close bound on the precision of a single volume probe, even if the sensor is allowed to separately optimize $\psi(\boldsymbol{r})$ in the interior and the exterior (Fig. 3a and Supplementary Note 10). In sum, we conclude that the optimal sensing strategy involves combining the results of multiple different measurements that probe the medium with different multipole symmetries.

**The precision of biomechanical sensing.** In this section, we extend our modeling framework to a scenario in which structural heterogeneity is known to play a significant role: cellular mechanosensing. Certain types of eukaryotic cells actively probe and respond to the stiffness of their surroundings, which has been shown to guide their behavior in decisive ways[24,56–60]. The importance of such mechanosensing invites the question of how precisely cells exploit the mechanical information available to them. In what follows, we present numerical evidence suggesting that some cells make optimal use of this information.

In connective tissue, a cell's mechanical environment primarily consists of a disordered biopolymer network that serves as a scaffold on which the cell lives and moves[28,29,61,62]. To quantify what a cell can learn by interacting with such a network, we generalize our sensing model to a three-dimensional, isotropic elastic medium characterized by a shear modulus $\mu$ and a Poisson's ratio $\sigma$ (Supplementary Note 11). For simplicity, we take $\sigma$ to be a fixed, uniform field and $\mu$ to be the sum of a fixed, uniform field $\mu_0$ and a spatially-varying random field $\delta\mu(\boldsymbol{r})$ with short-ranged correlations as in Eq. (3).

We determined the parameters in our model for a reconstituted collagen network, an in vitro system that closely resembles in vivo cellular environments[28,29,49,50]. For a collagen network prepared from a $c \sim 0.2$ μg/mL solution of collagen type-I monomers, previous studies suggest $\mu_0 \sim 0.3$ Pa, $\sigma \sim 0.4$, $\Delta_\mu \sim 0.1$ Pa$^2$, and $\xi \sim 5$ μm (Supplementary Note 11). For these values, the ratio $\Delta_\mu^{1/2}/\mu_0 \sim 1$ lies outside the strict regime of validity of our perturbative approach; nevertheless, we expect our results to qualitatively describe how $\delta\mu_0/\mu_0$ depends on the model parameters to the right order of magnitude.

Eukaryotic cells can sense stiffness by attaching to biopolymer networks via transmembrane protein complexes called focal adhesions[28,56,63,64]. For $D = 3$, the simplest cellular probe consists of isotropic dipolar shells of radius $a$ (Supplementary Note 11). Taking $a = 10$ μm leads to $\delta\mu_0/\mu_0 \sim 0.15$, which

supports the notion that cells could use mechanical information to reliably distinguish between different connective tissue environments, including brain ($\mu_0 \sim 1$ kPa), muscle ($\mu_0 \sim 10$ kPa), and bone ($\mu_0 \sim 100$ kPa)[28,57–60]. Such mechanosensing could be tested in experiment by using micropatterned materials to explore the effect of substrate heterogeneity on intracellular signaling dynamics. Interestingly, previous work in which cells were seeded on two-dimensional patterned substrates found that increases in either the average stiffness or in the spatial uniformity of substrate heterogeneities both resulted in increased intracellular signal transduction[27]. This agreement is consistent with Eq. (39), which suggests that increasing $\mu_0$ and decreasing $\xi$ should both have the same sign of influence on cellular precision.

A cell could further reduce $\delta\mu_0/\mu_0$ via sensory multiplexing. Indeed, cells have been known to modulate the forces they exert in order to vary the modes they apply, consistent with our predictions for behavior during sensory multiplexing[65]. A cell's spatial resolution is limited by the maximum number of focal adhesions that it can simultaneously apply to the network. Cells have been observed to display more than ~100 focal adhesions[66], which could allow a 10 μm cell to probe the network on scales smaller than $\xi$. Taking $d \sim \xi$ yields $\delta\mu_0/\mu_0 \sim 0.05$, which is comparable to the smallest relative differences in bulk stiffness that elicit significant changes in cellular motility and differentiation on homogeneous substrates[67]. This suggests that cellular mechanosensing may operate near the fundamental bounds on precision established in this paper.

An alternative strategy that cells could use to collect additional mechanical information is to perform multiple measurements by actively moving to different locations[67–69]. It is then natural to ask when this approach is preferable to remaining in one place and multiplexing probes with different symmetries. To determine the effectiveness of this strategy, we considered a cell that moves in a straight line and executes an isotropic dipolar probe every body length (Supplementary Note 12). For a 10 μm cell, we find that a cell must move about eight body lengths before $\delta\mu_0/\mu_0$ becomes smaller than 0.05, the corresponding value for a stationary cell from the previous paragraph. Thus, we expect that a cell would prefer to exert multiple probes in a single location if this could be done in less than the time it takes to move eight body lengths. Indeed, the precision of sensing improves more rapidly with the number of measurements for a stationary sensor than for a moving sensor (Supplementary Note 12). This suggests that cells should choose to stay put and probe their environment to the full extent permitted by their resolution $d$, after which they have no choice but to begin moving.

**The precision of active microrheology.** In this section, we apply our modeling framework to active microrheology in disordered polymeric systems and other soft materials. Active microrheology can be used for medical diagnostics, such as monitoring the progression of a cancer[52], and also for quality control during material fabrication[53,54]. Although previous studies have shown that active microrheology can be used to probe the properties of small systems[47], it has remained unclear what this technique can learn about heterogeneous materials. What is the fractional uncertainty of active microrheology, and what is the best strategy to estimate the average stiffness of a material using this approach?

To be concrete, we consider active microrheology that harnesses an engineered device to apply a static force and measure displacement at a single location inside an elastic medium (corresponding to the low-frequency limit of a general, frequency-dependent microrheology measurement). In contrast to our considerations regarding cells, such a device may exert a net force on the medium, e.g., by manipulating a bead using magnetic tweezers[47]. Thus, to explore what such a device can learn beyond what a cell is capable of discerning, we focus on a measurement protocol that consists of a monopole stimulus field and a monopole weight field applied to an elastic network. Such probe fields induce diverging deformations at the points of application. We account for these unphysical divergences by taking the measurement protocol to include a spherical cutoff of radius $\xi$ equal to the mesh size of the network (Supplementary Note 13). For the reconstituted collagen network we considered in the previous section, we used numerical integration to compute the fractional uncertainty of this measurement protocol and found $\delta\mu_0/\mu_0 \sim 0.2$. This value is small enough to reliably identify the presence of a cancer[70], which suggests that even one-particle active microrheology could be an effective tool for medical diagnostics.

Conventionally, active microrheology is done by probing the response of a material in a single direction. In a homogeneous material, this method yields results that do not depend on the direction of the probe. However, an anisotropic probe applied to a heterogeneous material generically yields a response that varies with the direction of the probe. In this case, a more precise measurement could be obtained by probing in different directions and combining the results via sensory multiplexing. Intuitively, the amount of information obtained by these probes may be maximized by spreading out their directions as much as possible. We found that $\delta\mu_0/\mu_0$ does indeed decrease with the number of samples taken, but with negligible improvements beyond $\delta\mu_0/\mu_0 \sim 0.15$ (Supplementary Note 13). The precision saturates to a non-zero value because monopoles couple to a fixed spatial extent of the medium regardless of orientation. Thus, the best strategy to probe $\mu_0$ by one-particle microrheology is to perform three probes in orthogonal directions, which allows a ~25% improvement in precision over a single monopole probe.

## Discussion
An understanding of fundamental bounds on sensing precision and information transmission has a long history of spurring advances in the sciences and engineering, ranging from improvements in telephony driven by Shannon's initial formulation of information theory to investigations of cell signaling and chemotaxis growing out of the Berg-Purcell limit on concentration sensing[40–42,71]. Studies of the limits of sensor performance are valuable both because they imply design constraints that engineered and evolved systems must satisfy and because determining the optimal performance often uncovers strategies to reach this optimum that can be used to improve performance even if the optimum cannot be attained.

Here, in the spirit of these earlier studies, we have quantified what a physical sensor can learn by probing a heterogeneous material. For media with long-ranged response functions, the smallest possible fractional uncertainty in estimating an average material constant is $\delta\lambda/\lambda_0 \sim (\Delta_\lambda^{1/2}/\lambda_0)(d/a)^{D/2}(\xi/a)^{D/2}$ for $a \gg \xi \gg d$, $\lambda_0 \gg \Delta_\lambda^{1/2}$, and $D > 1$. Remarkably, this relation implies that a finite-sized sensor applied to a standard elastic medium can achieve arbitrarily high precision—in effect, averaging the material constant field $\lambda(r)$ over an arbitrarily large volume—provided that it can perform multiple measurements down to small enough scales $d$. This "sensory multiplexing" provides a novel design principle for engineering high precision sensors that would be well-suited for applications on the microscopic scale[16–23,72].

In practical terms, our results imply that material properties can be estimated most precisely by making several measurements that each impose different multipole symmetries. Importantly,

one does not have to do a very large number of measurements to benefit from this strategy, as simply employing dipole and quadrupole probes can yield substantial improvements. These conclusions have implications both for the design of engineered sensors and for the behavior of living cells. In particular, cells can obtain additional information about the stiffness of their environment by applying multiple probes using forces that vary in space and time. Interestingly, such regulation of probes has been found to influence cellular differentiation[73], and cells in culture have recently been observed to vary the spatial symmetries of forces they exert[65].

For simplicity, we focused primarily on spherical sensors embedded inside a medium. However, our framework can also be used to study different sensory geometries and motile sensors. Many sensors operate on the boundary of media, including cells grown on flat surfaces[27]. Moreover, cells in connective tissue can become highly elongated[66] and undergo directed migration[74], both of which may serve as strategies for overcoming spatial correlations. Future experimental studies will be important to investigate the tradeoffs that cells employ between migrating and varying the angular distribution of their focal adhesions.

Throughout the main text, we assumed a vanishingly small material correlation length $\xi$, which holds provided that $d \gg \xi$. Our approach can be readily extended to account for a finite correlation length $\xi$ (Supplementary Note 14). Moreover, although we have focused mostly on a simple scalar version of elasticity, we expect our scaling results to hold for a broad range of media with long-ranged response functions, including the elastic medium in section "The precision of biomechanical sensing". For short-ranged response functions, the smallest possible fractional uncertainty is $\delta\lambda_0/\lambda_0 \sim (\Delta_\lambda^{1/2}/\lambda_0)(\xi/a)^{D/2}$ for $a \gg \xi$ and $\lambda_0 \gg \Delta_\lambda^{1/2}$, which can be interpreted as the familiar $1/\sqrt{N}$ scaling of measurement uncertainty for $N$ independent samples, where $N \sim (a/\xi)^D$ corresponds to the number of effectively independent subvolumes probed by the sensor (Supplementary Note 14). Finally, we assumed that the elastic properties of the medium within the sensing volume are not significantly mismatched from those of the exterior. Extending our model to account for more complicated constitutive relations and other distributions of the disorder are important directions for future research.

We have focused on athermal materials. For thermal materials, the quantities measured by the sensor fluctuate in time. These fluctuations provide an additional source of temporal noise to the inference process, as well as additional response configurations that can be observed by the sensor. Generalizing our approach to account for these effects would provide a comprehensive physical limit to sensing the properties of materials.

In summary, we have elucidated the perception of material properties in physical space. On small scales, structural heterogeneities place limits on the precision of sensing. We modeled these limits for biopolymer networks and found that they are comparable to the bounds observed for cells in experiment[67]. Going forward, our theory will guide the design of the next generation of sensors that will be capable of probing materials at the fundamental limits of spatial resolution.

## Methods

### Numerical minimization of $\delta\lambda_0/\lambda_0$.
To determine the optimal measurement protocol for a sensor that can apply arbitrary probe fields on its boundary, we used Mathematica's NMinimize function to search for a global minimum of $\delta\lambda_0/\lambda_0$. We performed this minimization over the coefficients $B_k^{(f)}$ and $B_k^{(w)}$ using the built-in Nelder–Mead method. The accuracy and precision goals were both chosen to be $\epsilon = 8$, and we took the maximum number of iterations to be $N_{max} = 1000$. To explore different local minima, we introduced stochasticity by repeating the minimization for 25 random initial seeds for each choice of the parameter $k_{max}$. For

each value of $k_{max}$, the minimum fractional uncertainty $\delta\lambda_{0,min}/\lambda_0$ reported was taken to be the minimum of the values found among the 25 trials.

## Data availability
The data that support the findings of this study are available from the corresponding author upon reasonable request.

## Code availability
We have made the Mathematica notebook used to minimize $\delta\lambda_0/\lambda_0$ available freely on GitHub (https://github.com/farzanb/sensing-in-random-media).

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

## Acknowledgements

We thank Aris Alexandradinata, Masud Beroz, William Bialek, Chase Broedersz, Judith Höller, David Huse, Tim (Hou Keong) Lou, Yigal Meir, Joshua Shaevitz, Ian Tobasco, and Ned Wingreen for insightful comments and discussions. This work was supported in part by the National Science Foundation Grants DMR-1056456 (to D.K.L.), DMR 1609051 (to X.M.), and EFRI-1741618 (to D.Z. and X.M.), a Margaret and Herman Sokol Faculty Award (to D.K.L.), and a Michigan Life Sciences fellowship (to F.B.).

## Author contributions

F.B. conceived the research. F.B. and D.Z. performed all simulations and analysis. All authors contributed to the interpretation of data and manuscript preparation.

## Competing interests

The authors declare no competing interests.
