## [Peer Review File · Nature Communications]

Reviewers' comments:

Reviewer #1 (Remarks to the Author):

The manuscript at hand explores the linear response to multiple stimuli of a quenched disordered elastic medium. If the stimuli are sensors, then the authors demonstrate, for example, that a finite-sized sensor can attain arbitrarily high-precision for multiple measurements at some small enough length scale. Presumably, this manuscript is a somewhat natural consequence of an earlier publication by the first author (and others) on mechanosensing in disordered fiber networks. In particular, in this initial work, the effect of a dipole in a continuum version of an elastic medium with the effects of fiber stiffness incorporated, etc. Presumably, the focus on the effect of a dipole is motivated by thinking of the cell as a mechanosensor. In the manuscript at hand, the mechanosensing aspect of the cell is considered to be the focal adhesions, with each making its own measurement, as opposed to the overall contractility of the cell as presented in the initial work (hence the focus on the dipole).

I remain on the fence about this manuscript because it seems that while the authors have derived a new limit on the sensory capability of multiple probes in a Gaussian-disordered elastic medium, the notion of arranging multiple probes in such a way so as to overcome interference effects (such as pairing each probe with a companion probe) to attain this limit seems a bit "forced". In addition, as pointed out in Reference 13, T. Lubensky and B. DiDonna demonstrate that there will be local deviations from affinity in disordered elastic media, which the authors do not appear to address in their formalism, and, would, in principle, mean that one must go beyond a mesoscopic generalization of Hooke's law. Finally, while the authors apply their formalism to cellular mechanosensing (in Section 5), I am not necessarily convinced that the number and size of focal adhesions in a cell are "aware" of the new limit derived by the authors. Given the amount of already available data on focal adhesions, etc. in differing mechanical environments, perhaps the authors could make a stronger case for the application of their ideas to cells.

Reviewer #2 (Remarks to the Author):

The authors present a study of the limits of resolution of a scalar quantity using finite area probes. The paper is an original approach to a general problem and deserves publication but the results, which are quite formal and without any strong specific application - at least in this paper, do not make a strong case for publication in a journal of general significance like Nature Comm. Furthermore, the paper, as written, does not present a clear account of either the motivation for the model or its analysis. I cannot recommend publication in Nature Comm.

Specific comments:

- 1) A length scale is introduced in Eq.2 even though a λ is δ correlated. In what sense is it a length when there is no actual non-local correlation?
- 2) Given its importance, it would help if the role of the ψ field defined in Eq. 7 was explained.
- 3) Some explanation should be provided for why the analysis in Sect. III is for the boundary of the probe area - why not probe the entire area? Also - the motivation for the complex structure of the probe fields is not provided. Is there really any likelihood of probes access much more than a quadrupole distribution?
- 4) The decision to neglect the $k=0$ term (p. 3) comes as a surprise since it seems the most obvious choice. The justification "to avoid pathological non-decaying interactions of monopoles" makes no sense whatsoever and then on p. 5 the $k=0$ term is included anyway.
- 5) The fields $\psi_+(r)$ and $\psi_-(r)$ were not explained.
- 6) Much of p.4 is spent discussing behavior for large k_{\max} without ever establishing why this is likely to be of any practical interest.
- 7) The term "multiplexing" is really not that informative. If I have understood it means using separate measurements each characterized by a single harmonic as opposed to a single measurement constructed from the sum of the harmonics. Again - the reader is not really supplied with enough information to decide whether this is particularly important or not.
- 8) The application in Sect. V does not provide much additional insight. If this is a reasonable example of the kind of application of this method then I really don't understand why it shouldn't appear in some more specialist journal.

Reviewer #3 (Remarks to the Author):

The manuscript "Physical limits to sensing material properties" by Farzan Beroz, Di Zhou, Xiaoming Mao, and David Lubensky address a fundamental problem of broad interest. Given the increasing precision of modern measurement and sensing techniques, it is a timely topic.

The paper directly motivates improved experimental designs based on a rigorous mathematical analysis with reasonable assumptions. It is based on a good choice of methods.

Besides the importance to applications, the paper offers interesting analytic deductions, fascinating questions and connections to other fields of research.

My suggestions and questions below mainly address the presentation, which is currently quite technical and brief. For a non-expert, the nice results are therefore hard or impossible to understand. Reading the arxiv version helped me a lot to understand the paper. Some of its nice intuitive explanations should be included in this manuscript even if the length-constraints pose a problem. So, it seems to me that all my questions and the problem of being readable for a broad audience can be solved.

Conditional to these clarifications, I recommend publication in Nature Communications.

The Supplementary Information contains extended calculations and derivations, which in the given time, I could only check randomly. The paper is carefully written, for example, with hardly any typos.

Suggestions and questions for clarification:

- * By what argument is the analysis in Secs. IV and V restricted to a ring? Is that a limitation or assumption that should be pointed out more clearly, or is there an argument why this is an optimal choice? For example, in Sec. IV, it says in the last paragraph: "even if the sensor is permitted to apply an arbitrary pair of probe fields within its volume".
 - * Is it guaranteed that the optimal probe fields at high k_{\max} are nonisotropic? Could this be just deep local minima found by the repeated minimization with a Nelder-Mead algorithm? What is the dimension of the parameter space? Why was the latter chosen and not, e.g., simulated annealing which is implemented in Mathematica, too?
 - * At first reading, it is confusing to see Eq. (27) stated for general D if in the next line it is restricted to $D=2$. Only in the next paragraph, it is stated that it also holds for $D=3$. The result is stated for $D>1$. Has it been proven for any dimension or does this refer to $D=2$ and $D=3$?
 - * In Sec. V, I appreciate the interesting summary of cellular mechanosensing and the prediction based on parameters measured in previous studies. There are some interesting pointers to future research, in particular, how the theory can be tested in experiment.
- Just a few questions for clarification:
- * The sentence "we consider our sensing model for a three-dimensional" should actually say the the model is generalize to three dimensions, shouldn't it? This is announced in the introduction and carried out in a quite extended calculation in the Supplementary Note. Is that correct? *
 - * What exactly represents the elastic sheet, that is, what is the relation between the collagen network with $\mu_0 \sim 0.3 \text{ Pa}$ and bone with $\mu_0 \sim 100 \text{ kPa}$?
 - * Do I understand correctly that all three assumptions of the perturbative approach, $\Delta \ll \mu_0^2$ and $\xi \ll d \ll a$, are not fulfilled?
 - * To what does 'qualitatively describe' refer in the third paragraph? Does it refer to the right order of magnitude, which would be sufficient for the examples given?
 - * Are there less extreme examples, for which the actual limit could be tested experimentally?

Minor questions and suggestions:

- * The formula in the abstract clearly states the limits of its validity. For clarity, I suggest to already mention in the second sentence that the analysis is based on a perturbative approach.
- * I do not quite get the motivation for the energy in Eq. (1). Is it related to the Helfrich Hamiltonian up to leading order?
- * The restriction to a Gaussian random field is uncritical (basically due to the central limit

theorem). Nevertheless, why is it necessary if we anyhow assume a vanishing correlation length?

- * Does "nonlocal" mean unbounded support?

- * An explicit definition of $\delta \lambda_0$ as the standard deviation might avoid it to be misinterpreted as a bias (which would be zero).

- * Why are some of the configurations generated by Eq. (25) unphysical, if the additional Eq. (28) is only implied by the choice of normalization?

- * Just an optional suggestion: Without explicit calculations just mentioning the measurement procedure, can the examples man-made sensors in the introduction (or at least one of these) be directly related to the model? If this can be done already in or before Sec. II, this might increase the readability for non-experts.

Reviewers' comments:

Reviewer #1 (Remarks to the Author):

The manuscript at hand explores the linear response to multiple stimuli of a quenched disordered elastic medium. If the stimuli are sensors, then the authors demonstrate, for example, that a finite-sized sensor can attain arbitrarily high-precision for multiple measurements at some small enough length scale. Presumably, this manuscript is a somewhat natural consequence of an earlier publication by the first author (and others) on mechanosensing in disordered fiber networks. In particular, in this initial work, the effect of a dipole in a continuum version of an elastic medium with the effects of fiber stiffness incorporated, etc. Presumably, the focus on the effect of a dipole is motivated by thinking of the cell as a mechanosensor.

We thank the reviewer for noting that the first author previously published a manuscript on mechanosensing in disordered fiber networks. In the earlier publication, the author presented modeled distributions of local stiffnesses measured by dipolar sensors applied to elastic networks. These elastic networks were not continuum versions of elastic media with fiber stiffness incorporated, but instead were collections of discrete fibers modeled as spring-like elements. The central conclusion of this work was that the local stiffness distributions for fiber networks are generically very broad.

In contrast to this early modeling work, the current manuscript under consideration answers an entirely different question. Rather than exploring how the local stiffness distribution is affected by a material's microstructure, here we ask, given a stiffness distribution, what ultimately limits a probe's ability to infer the average stiffness? These results apply to any elastic materials, not just to fiber networks. We have underlined this point by introducing an additional section (Section VII) that applies our formalism to active microrheology of any soft materials.

In the manuscript at hand, the mechanosensing aspect of the cell is considered to be the focal adhesions, with each making its own measurement, as opposed to the overall contractility of the cell as presented in the initial work (hence the focus on the dipole).

We agree that the type of mechanical sensing we consider in the manuscript at hand includes sensing that may be performed via focal adhesions. The formalism in the current manuscript allows for any pattern of forces to be applied to the focal adhesions, all of which would ultimately arise from active contractions within the cell. This includes, in particular, isotropic contractions and force dipoles. We have clarified this point by modifying the first sentence of the fourth paragraph of Section VI.

I remain on the fence about this manuscript because it seems that while the authors have derived a new limit on the sensory capability of multiple probes in a Gaussian-disordered elastic medium, the notion of arranging multiple probes in such a way so as to overcome interference effects (such as pairing each probe with a companion probe) to attain this limit seems a bit "forced".

We thank the reviewer for underlining the novelty of our results on the sensory capability of multiple probes. As regards the concern that some of our conclusions might be “forced,” we would first like to emphasize that our basic theoretical framework is not limited to biological applications. Whereas we agree that it is unlikely that a living cell will apply a precisely optimal arrangement of probes that reaches the fundamental bound on sensor performance, it is much easier to imagine that something close to the optimum could be achieved in a human-engineered system. To illustrate the broader applicability of our findings, we now include a brief section (Section VII) on their implications for microrheology experiments.

Turning specifically to the biological relevance of our results, it is worth taking a step back to consider why calculating fundamental limits on sensing and information transmission has been such a powerful approach in many areas of biological physics (as exemplified, e.g., by the Berg-Purcell limit on receptor performance or the work of Bialek and others on channel capacities in neural information processing). We would suggest that there are two reasons.

First, it is useful to know whether the measured performance of a biological system is close to its fundamental physical limits. If it is, this usually constrains system architecture in ways that are both predictive and valuable for understanding the biological function of otherwise mysterious features. Here, we have derived the fundamental performance bounds for a new class of sensing problems of considerable biological importance; this is a necessary prerequisite to exploring how close to these bounds living systems actually operate.

Second, solving an optimization problem is a theoretical approach that allows one to discover, in a systematic way, strategies that can be used to improve performance even fairly far from the optimum. For the mechanosensing problem that we study, two such strategies stand out: 1) Combining the estimates from more than one independent probe – that is, sensory multiplexing – can greatly improve the accuracy of inferred elastic moduli. Thus, we predict that cells that need to determine the elastic properties of their environment should vary the pattern of forces they exert over time. 2) More specifically, we find that the largest improvement in accuracy is obtained by combining probes corresponding to different multipoles. Thus, we expect that cells that are measuring elastic moduli should not always remain strongly polarized, as this would limit them to only dipole probes. Instead, their loading patterns should sometimes be dipolar and at other times have quadrupolar or higher symmetries. Interestingly, we have identified an experimental study that observed that cells modulate the forces they exert in such a manner (see Section VI). We have added language to our manuscript to highlight these clear, qualitative conclusions of our formalism.

Finally, we note that our results do not rely on the one particular arrangement of probes that we consider in Section V. The ability to extract additional information using multiple probes requires only that the different probes yield information about different regions of the medium. To illustrate this point, we have described another arrangement of probes in Section 6 of the Supplementary Information that achieves the same limit using a different number of probes (compare equation S109 to equation S111). The ability to extract additional information using multiple probes requires only that the different probes yield information about different regions of the medium. We have

clarified this point by providing more detail in the second sentence following equation 36.

In addition, as pointed out in Reference 13, T. Lubensky and B. DiDonna demonstrate that there will be local deviations from affinity in disordered elastic media, which the authors do not appear to address in their formalism, and, would, in principle, mean that one must go beyond a mesoscopic generalization of Hooke's law.

We agree that there will be local deviations from affinity in disordered elastic media. We have addressed these deviations in our formalism. Indeed, we have used the same formalism as T. Lubensky and B. DiDonna in the absence of random prestress in the elastic media (compare equation 2.15 and equation 3.25 in Reference 13 to equation 15 in our manuscript, which agrees with these equations in the absence of prestress).

Finally, while the authors apply their formalism to cellular mechanosensing (in Section 5), I am not necessarily convinced that the number and size of focal adhesions in a cell are "aware" of the new limit derived by the authors. Given the amount of already available data on focal adhesions, etc. in differing mechanical environments, perhaps the authors could make a stronger case for the application of their ideas to cells.

We thank the reviewer for this intriguing suggestion, and we have included additional references to studies that connect our ideas to cells in Section VI. We would like to emphasize an experiment by Hadden et al. (2017) we referenced in the fifth paragraph of Section VI, which found that cells respond to differences in bulk stiffness down to approximately 5% on homogeneous substrates. The value of 5% corresponds to our limit of cellular precision for sensory multiplexing, and is smaller than the 15% limit of precision for a cell in a typical, highly disordered biopolymer network without sensory multiplexing. This observation suggests that the number and size of focal adhesions may have evolved to approach the limit we derived in natural heterogeneous environments.

Most previous work on cellular mechanosensing has employed homogeneous environments. Interestingly, however, a relevant study by Yang et al. (2016) has examined cells grown on patterned heterogeneous substrates. The changes observed in intracellular signaling with variations in these substrates' properties are consistent with the limit we derived. We now discuss this observation in the fourth paragraph of Section VI.

Finally, we have identified several previous studies that support the application of our results to cells. A previous study by Hu et al. (2017) observed an influence of focal adhesion turnover on cellular differentiation. Our formalism suggests that such focal adhesion dynamics could allow the cells to perform multiple probes of stiffness. We have highlighted this new prediction and its experimental support in a new paragraph in the Discussion. In addition, a previous study by Messi, Bornert, Raynaud, and Verkhovsky (2019) observed cells varying the forces they exert to apply different angular distributions of forces in time, which is a hallmark of the sensory multiplexing strategy we proposed. We now cite this observation in the fifth paragraph of Section VI.

(See also our discussion above of qualitative predictions for cells, in response to one of the referee's earlier points.)

Aside from the applications of our formalism to cellular mechanosensing, our results provide the ultimate limits to sensing the properties of materials using any approach, as well as strategies for how to operate at these limits. These strategies may be used with engineered sensors, e.g. in microrheology, which we now treat in Section VII of our updated manuscript.

Reviewer #2 (Remarks to the Author):

The authors present a study of the limits of resolution of a scalar quantity using finite area probes. The paper is an original approach to a general problem and deserves publication but the results, which are quite formal and without any strong specific application - at least in this paper, do not make a strong case for publication in a journal of general significance like Nature Comm.

We thank the reviewer for commenting on the originality and generality of our approach, and for suggesting that our paper could benefit from the addition of strong specific applications. Although we had already included an application of our results to cellular mechanosensing, our findings also apply more broadly to other types of biological sensors and to engineered sensors. To address this point, we have expanded our discussion of the specific implications of our formalism for cell behavior, and we have added a new section (Section VII) which treats microrheology as an example of a non-biological application of our formalism. We believe that these examples serve to clearly illustrate the general significance of our findings.

Furthermore, the paper, as written, does not present a clear account of either the motivation for the model or its analysis. I cannot recommend publication in Nature Comm.

We have clarified the motivation of the model and its analysis by identifying additional experiments in agreement with our results (Section VI), providing new predictions that could be investigated in future experiments (Section VI and Discussion), and also by including an additional section applying our results to microrheology (Section VII). The third paragraph of the Introduction now explicitly states the motivation for the model and its analysis. In particular, we note that the precision of probing heterogeneous materials has remained unknown before going on to state the main questions we address, which are to determine the limits of sensing the properties of heterogeneous materials and determining how a physical device can be designed to achieve these limits. Our results bear on understanding cellular behavior, engineering sensors, and improving material fabrication.

Specific comments:

1) A length scale is introduced in Eq.2 even though a λ is δ correlated. In what sense is it a length when there is no actual non-local correlation?

The referee is correct that in the limit that the disorder is delta-correlated, the introduction of a length scale ξ amounts to a somewhat arbitrary parameterization choice that aids in dimensional analysis. We have, however, chosen to express the noise correlator this way so that it is possible to smoothly connect our results for the delta correlated case to the more general problem of spatial randomness with a finite correlation length. To clarify this point, we have now included an additional equation (Eq. 2 in the resubmitted version) prior to this equation that describes situations where ξ is not much smaller than the sensor resolution d .

2) Given its importance, it would help if the role of the ψ field defined in Eq. 7 was explained.

We thank the reviewer for this helpful suggestion. We now define the role of the field ψ upon introducing it.

3) Some explanation should be provided for why the analysis in Sect. III is for the boundary of the probe area - why not probe the entire area? Also - the motivation for the complex structure of the probe fields is not provided. Is there really any likelihood of probes access much more than a quadrupole distribution?

The analysis in Section IV (formerly Section III) treats the boundary of the probe area, and later on in our manuscript we treat the probe of the entire area. In the second sentence of Section IV, we have clarified that we begin by considering the similar problem of probes on the boundary of the probe area. Importantly, the effects of the probe fields extend well beyond the probe area, because of the long-ranged nature of elastic responses. Thus, the difference between the two cases is not as large as it might naively seem.

Our motivation for introducing the particular probe fields used in our paper is to determine a fully general limit to the precision of sensing. Importantly, this limit bounds the precision of any sensor, regardless of the specific probe fields employed by the sensor. Moreover, as we argue in more depth above in our response to Reviewer #1 (paragraph beginning "Turning specifically to the biological relevance of our results..."), it is of value to know the fundamental bound even if in some (e.g. biological) cases it seems unlikely that real sensors will be able to perform the complicated probes needed to saturate that bound. It is also possible to engineer sensors that apply the probe fields used in our analysis.

4) The decision to neglect the $k=0$ term (p. 3) comes as a surprise since it seems the most obvious choice. The justification "to avoid pathological non-decaying interactions of monopoles" makes no sense whatsoever and then on p. 5 the $k=0$ term is included anyway.

Applying a monopolar stimulus to a two-dimensional elastic medium causes displacements with amplitudes that grow logarithmically with the distance from the point of application; this divergence leads to pathological behavior within our formalism. Of

course, in reality no physical system is ever truly two-dimensional, and the displacement field from a monopolar force must always eventually cross over to a three-dimensional, $1/r$ decay. Dealing with the mathematical complexities of this crossover is beyond the scope of this initial paper. More importantly, there is a large class of physical situations where it is justified – in fact, necessary – to neglect “ $k=0$ ” modes. Any sensor, like a living cell, that is not subject to externally applied body forces cannot, by Newton’s 3rd law, exert a net force on the medium. We have now clarified this point in the manuscript text.

We have not included the monopolar stimulus in Section V (formerly on p. 5). However, we are grateful that the reviewer has pointed out that writing that we have used a separate term with “ $k=0$ ” may cause confusion. To address this point, we have modified the sentence in Section V with “ $k=0$ ” to include the word “companion”.

5) The fields $\psi_+(r)$ and $\psi_-(r)$ were not explained.

We have clarified the sentence after these fields are introduced to explain that these fields are probe intensities that correspond to the interior and exterior of the sensor.

6) Much of p.4 is spent discussing behavior for large k_{max} without ever establishing why this is likely to be of any practical interest.

We discuss the behavior for large k_{max} to establish a limit to the precision of any sensor. This limit addresses an important hypothesis of our work, which states that cells may operate near the limits of sensing. As discussed above (under point 3 of Reviewer #2 and in the response to Reviewer #1), it has proven valuable to know the fundamental limits on sensing precision even when particular biological systems do not reach these limits. In addition, we point out in the Discussion that it is possible to engineer artificial sensors that can apply probe fields with large k_{max} , which would be useful for medical applications.

7) The term "multiplexing" is really not that informative. If I have understood it means using separate measurements each characterized by a single harmonic as opposed to a single measurement constructed from the sum of the harmonics. Again - the reader is not really supplied with enough information to decide whether this is particularly important or not.

The referee’s understanding of our use of the term is correct. We have now removed the term “sensory multiplexing” from the Introduction. We now use the term only after defining it in Section V. Moreover, we explicitly state that the sensory multiplexing strategy would allow sensors to gain a significant amount of additional information.

8) The application in Sect. V does not provide much additional insight. If this is a reasonable example of the kind of application of this method then I really don't understand why it shouldn't appear in some more specialist journal.

Section VI (formerly Section V) directly addresses cellular mechanosensing, which is an important experimental system with broad medical applications. These results identify tradeoffs that occur for different ways that cells may vary their focal adhesions. We now highlight these tradeoffs in the second paragraph of the Discussion. Finally, the application in Section VI provides additional insight by demonstrating how our framework describes standard three-dimensional elastic media, which are the most common types of materials.

Aside from the application to cellular mechanosensing, our method also describes sensing performed by engineered sensors. To that end, we have used our formalism to address microrheology (see section VII). Based on the broad applicability of our results to mechanical sensing and to other types of sensing, we believe that our approach deserves publication in *Nature Communications*.

Reviewer #3 (Remarks to the Author):

The manuscript "Physical limits to sensing material properties" by Farzan Beroz, Di Zhou, Xiaoming Mao, and David Lubensky address a fundamental problem of broad interest. Given the increasing precision of modern measurement and sensing techniques, it is a timely topic.

The paper directly motivates improved experimental designs based on a rigorous mathematical analysis with reasonable assumptions. It is based on a good choice of methods.

Besides the importance to applications, the paper offers interesting analytic deductions, fascinating questions and connections to other fields of research.

We thank the reviewer for these encouraging comments.

My suggestions and questions below mainly address the presentation, which is currently quite technical and brief. For a non-expert, the nice results are therefore hard or impossible to understand. Reading the arxiv version helped me a lot to understand the paper. Some of its nice intuitive explanations should be included in this manuscript even if the length-constraints pose a problem. So, it seems to me that all my questions and the problem of being readable for a broad audience can be solved.

Conditional to these clarifications, I recommend publication in *Nature Communications*.

We are grateful for these positive comments on the arxiv version. We now include the results mentioned in the arxiv version. In particular, we now treat an elastic medium for which nearby points are not coupled to each other. In addition, we have included some other more intuitive explanations throughout the updated version of our manuscript.

The Supplementary Information contains extended calculations and derivations, which in the given time, I could only check randomly. The paper is carefully written, for example, with hardly any typos.

Suggestions and questions for clarification:

* By what argument is the analysis in Secs. IV and V restricted to a ring? Is that a limitation or assumption that should be pointed out more clearly, or is there an argument why this is an optimal choice? For example, in Sec. IV, it says in the last paragraph: "even if the sensor is permitted to apply an arbitrary pair of probe fields within its volume".

The analysis in Sections V and VI (formerly Sections IV and V) is not restricted to a ring. The sentence quoted from Section V refers to the analysis in Section IV performed for probes restricted to a ring. In the updated version of our manuscript, we have clarified that the restriction in Section IV is performed for simplicity.

* Is it guaranteed that the optimal probe fields at high k_{\max} are nonisotropic? Could this be just deep local minima found by the repeated minimization with a Nelder-Mead algorithm? What is the dimension of the parameter space? Why was the latter chosen and not, e.g., simulated annealing which is implemented in Mathematica, too?

The optimal probe fields are guaranteed to be nonisotropic. To demonstrate this, we have analytically determined the optimal isotropic probe fields and found nonisotropic probe fields that perform more precisely (see Fig. 3a).

While working on the manuscript, we first used all of the algorithms in Mathematica, including the simulated annealing algorithm, for smaller values of k_{\max} . However, we eventually used the Nelder-Mead algorithm for our final analysis because it converged more quickly for larger values of k_{\max} .

* At first reading, it is confusing to see Eq. (27) stated for general D if in the next line it is restricted to $D=2$. Only in the next paragraph, it is stated that it also holds for $D=3$. The result is stated for $D>1$. Has it been proven for any dimension or does this refer to $D=2$ and $D=3$?

The result has been proven for $D=2$ and $D=3$. We have modified the paragraph pointed out by the reviewer to clarify that our results refer to $D=2$ and $D=3$.

* In Sec. V, I appreciate the interesting summary of cellular mechanosensing and the prediction based on parameters measured in previous studies. There are some interesting pointers to future research, in particular, how the theory can be tested in experiment.

Just a few questions for clarification:

* The sentence "we consider our sensing model for a three-dimensional" should actually say the the model is generalize to three dimensions, shouldn't it? This is announced in the introduction and carried out in a quite extended calculation in the Supplementary Note. Is that correct?

We thank the reviewer for mentioning their appreciation. We have revised the noted sentence to say that we generalize the model to three dimensions. It is correct that we carry out the calculation in the Supplementary Note.

* What exactly represents the elastic sheet, that is, what is the relation between the collagen network with $\mu_0 \sim 0.3$ Pa and bone with $\mu_0 \sim 100$ kPa?

The value “ $\mu_0 \sim 0.3$ Pa” was quoted for a three-dimensional *in vitro* system used in a previous study and in the current study. The stiffnesses of crosslinked gels can vary by orders of magnitude depending on parameters like the crosslink density, and this stiffness lies towards the lower end of the biologically relevant range for collagen gels. Nonetheless, we chose to focus on this system because it is a well-characterized model for connective tissue, for which the heterogeneous structure has been directly observed and measured.

The values 1 kPa, 10 kPa, and 100 kPa are for *in vivo* connective tissue environments on larger scales and include contributions from living cells and other components beyond simply collagen.

Various experimental models of cell motion and differentiation in culture employ environments spanning the entire range of stiffnesses, from \sim Pa to \sim 100 kPa, and make strong cases that these environments are relevant to different processes in living systems. Thus, different cell types, in different contexts, might plausibly need to sense stiffnesses across this entire range. The machinery of biology appears to be able to adjust to this challenge, and we do not seek to address here the mechanistic details of how this is accomplished. Instead, we are interested in the fundamental limits on sensing precision imposed by the physics of mechanical response, under the assumption that cells can detect local displacements u arbitrarily well. Importantly, our formalism to describe these limits does not depend on absolute stiffness values, only on relative variations.

* Do I understand correctly that all three assumptions of the perturbative approach, $\Delta \mu_0^2$ and $\xi \ll a$, are not fulfilled?

Our analysis of the results from a previous study (Ref. 29) to fit the parameters of our model suggests that these assumptions are not strictly fulfilled for the specific case of the model *in vitro* collagen network. Nevertheless, we expect our results to qualitatively describe how the precision of sensing depends on the model parameters, which we have noted in Section VI.

* To what does 'qualitatively describe' refer in the third paragraph? Does it refer to the right order of magnitude, which would be sufficient for the examples given?

These words refer to the right order of magnitude and to the right trends as different parameters are varied. We have updated the paragraph to include this clarifying statement.

* Are there less extreme examples, for which the actual limit could be tested experimentally?

We have proven that the actual limit holds for all examples of media, including less extreme examples than the fibrous networks of connective tissue. Thus, the limits could be tested for an *in vitro* system that contains heterogeneities on scales larger than those of connective tissue.

Minor questions and suggestions:

* The formula in the abstract clearly states the limits of its validity. For clarity, I suggest to already mention in the second sentence that the analysis is based on a perturbative approach.

We thank the reviewer for this clarifying suggestion. We have modified the second sentence of the abstract to state that the analysis is based on a perturbative approach.

* I do not quite get the motivation for the energy in Eq. (1). Is it related to the Helfrich Hamiltonian up to leading order?

The energy in Eq. 15 (formerly Eq. 1) is chosen because it is a general and simple model for an elastic medium with spatially-varying elastic properties.

* The restriction to a Gaussian random field is uncritical (basically due to the central limit theorem). Nevertheless, why is it necessary if we anyhow assume a vanishing correlation length?

We agree that the restriction to a Gaussian random field is uncritical and unnecessary. We have removed this restriction from the manuscript.

* Does "nonlocal" mean unbounded support?

We use the term "nonlocal" to refer to a function at a given point that depends on at least one value at a different point in space.

* An explicit definition of $\delta \lambda_0$ as the standard deviation might avoid it to be misinterpreted as a bias (which would be zero).

We thank the reviewer for this helpful suggestion. We now define $\delta \lambda_0$ as the standard deviation in the last paragraph of Section III.

* Why are some of the configurations generated by Eq. (25) unphysical, if the additional Eq. (28) is only implied by the choice of normalization?

The configurations generated by equation 25 are all physical. We did not include an equation 28 in the original version of the manuscript, so we are somewhat confused what equation the referee intended to refer to.

* Just an optional suggestion: Without explicit calculations just mentioning the measurement procedure, can the examples man-made sensors in the introduction (or at least one of these) be directly related to the model? If this can be done already in or before Sec. II, this might increase the readability for non-experts.

We thank the reviewer for this helpful suggestion. We now note in the Introduction that our model directly relates to the examples of man-made sensors, and that we address these man-made sensors in Section VII.

REVIEWERS' COMMENTS:

Reviewer #2 (Remarks to the Author):

The authors have addressed my concerns regarding the original manuscript and I recommend that their revised paper be accepted for publication.

Reviewer #3 (Remarks to the Author):

The authors have thoughtfully and convincingly answered all of my questions. The manuscript now provides intuitive explanations and a promising discussion of relevant applications, which motivates further research.

I affirm my positive original review and recommend publication in Nature Communications.

Reviewers' comments:

Reviewer #2 (Remarks to the Author):

The authors have addressed my concerns regarding the original manuscript and I recommend that their revised paper be accepted for publication.

We thank the reviewer for their recommendation.

Reviewer #3 (Remarks to the Author):

The authors have thoughtfully and convincingly answered all of my questions. The manuscript now provides intuitive explanations and a promising discussion of relevant applications, which motivates further research.

I affirm my positive original review and recommend publication in Nature Communications.

We thank the reviewer for their positive comments and for affirming their recommendation.